# Epitaxial bulk acoustic wave resonators as highly coherent multi-phonon sources for quantum acoustodynamics

Vikrant J. Gokhale [1✉], Brian P. Downey[2✉], D. Scott Katzer[2], Neeraj Nepal[2], Andrew C. Lang[3], Rhonda M. Stroud [2] & David J. Meyer[2]

Solid-state quantum acoustodynamic (QAD) systems provide a compact platform for quantum information storage and processing by coupling acoustic phonon sources with superconducting or spin qubits. The multi-mode composite high-overtone bulk acoustic wave resonator (HBAR) is a popular phonon source well suited for QAD. However, scattering from defects, grain boundaries, and interfacial/surface roughness in the composite transducer severely limits the phonon relaxation time in sputter-deposited devices. Here, we grow an epitaxial-HBAR, consisting of a metallic NbN bottom electrode and a piezoelectric GaN film on a SiC substrate. The acoustic impedance-matched epi-HBAR has a power injection efficiency >99% from transducer to phonon cavity. The smooth interfaces and low defect density reduce phonon losses, yielding ($f \times Q$) and phonon lifetimes up to $1.36 \times 10^{17}$ Hz and 500 μs respectively. The GaN/NbN/SiC epi-HBAR is an electrically actuated, multi-mode phonon source that can be directly interfaced with NbN-based superconducting qubits or SiC-based spin qubits.

[1] National Research Council Fellow residing at the US Naval Research Laboratory, Washington, DC, USA. [2] US Naval Research Laboratory, Washington, DC, USA. [3] American Society for Engineering Education Postdoctoral Fellow residing at the US Naval Research Laboratory, Washington, DC, USA.
✉email: vikrant.gokhale.ctr.in@nrl.navy.mil; brian.downey@nrl.navy.mil

Quantum technologies for computation and sensing rely on the efficient interaction of a quantum state with a pump or probe signal. While fundamental physics investigations have focused on natural candidates that exhibit quantum states, such as trapped atoms or ions[1], practical quantum technology platforms require alternatives that are scalable. Widely researched cavity quantum electrodynamics (QED) strategies use the coupling between qubits and microwave or optical photons trapped in a resonant cavity[1]. In order to reach the strong coupling regime between photon and qubit, and to reduce decoherence, both the qubit and the cavity should have extremely low levels of dissipation. This requires a high quality factor ($Q$) photonic cavity such as a Fabry–Perot cavity for optical photons, or a superconducting transmission line cavity resonator for microwave photons. An analogous approach, cavity quantum acoustodynamics (QA or QAD)[2–5], couples qubits with cavity phonon modes. The use of phonons has a crucial advantage: phonons in crystal lattices have a velocity slower than electromagnetic waves by a factor of $\sim 10^5$. The slower velocity enables quantum states (thus quantum information) to be preserved for a significantly longer time; as long as the phonon cavity has a high $Q$. The use of microscale acoustic transducers as phonon sources and cavities enables chip-scale fabrication and electrical drive/sense/control systems, significantly reducing footprint, power consumption, and complexity, as compared with off-chip macroscale phonon sources such as optomechanically transduced quartz resonators[6,7]. These factors become especially critical as we attempt to scale from single qubit–phonon interactions to arrays of qubit–phonon coupled devices[8,9]. Microfabricated acoustic transducers, including flexural beam resonators, surface acoustic wave (SAW) and bulk acoustic wave (BAW) cavity resonators can be easily adapted for use as phonon sources in QAD systems. Various combinations of materials, transducers, phonon modes, and qubit types have been successfully demonstrated in recent studies;[2–5,8–16] of these the high-overtone bulk acoustic resonator (HBAR) (also known as an overtone or composite BAW resonator) is a prime candidate as a compact microfabricated low-loss phonon source. The HBAR is a simple and robust BAW resonator conventionally made by sputter depositing a transducer comprised of a piezoelectric film sandwiched between metal electrodes onto a thicker low-loss substrate (in this context, a substrate that has very low intrinsic phonon loss). HBAR variants with curved surfaces for focusing acoustic waves[6,16,17], or in-plane phonon cavities for flexible design[18,19] have been investigated for various applications. HBAR studies have traditionally focused on applications in radio frequency (RF) signal processing, sensing, and spectroscopy[15,20–22]. Recently, HBARs have seen renewed interest as phonon sources in QAD experiments[5,10–13]. Achieving a high cavity $Q$ (and consequently increasing phonon lifetime and coherence) requires the systematic elimination or reduction of various energy dissipation mechanisms in the structure of the HBAR. The fundamental lower limit for acoustic energy dissipation is the anharmonic dissipation in the phonon cavity[23], but is generally overshadowed by phonon scattering at grain boundaries, point or line defects, phase impurities, inclusions, and rough surfaces and interfaces. To systematically reduce net loss in the phonon cavity, it is necessary to reduce (if not eliminate) defect-based and interface-based phonon scattering. Many QAD experiments use bulk single-crystal overtone BAW resonators: massive macroscale devices with no internal interfaces and very low defect density[6,7,17,24,25]. However, such devices often require optomechanical actuation and are difficult to integrate into a compact system. Most thin film HBARs demonstrated to date have consisted of evaporated/sputtered polycrystalline metal electrodes and sputter-deposited polycrystalline piezoelectric thin films on low-loss substrates such as sapphire, diamond, and SiC[3–5,10–16,19–21,26–29]. While the substrate makes up the largest fraction of the device, scattering losses in the piezoelectric thin film, the metal electrodes, and crucially, the interfaces between all these layers limit device performance[16,26,30]. We believe it is necessary, but not sufficient, to use a low-loss substrate when making a thin film HBAR device appropriate for use in QAD systems. For further improvement, the quality of the metal electrodes and piezoelectric film must be improved in order to achieve low dissipation that approaches the intrinsic material limits of the substrate.

This article presents epitaxially grown HBARs (epi-HBARs) with materials carefully chosen to provide acoustic impedance matching that results in a power injection efficiency >99%. The epi-HBARs consist of a transition metal nitride (TMN)/ group III-nitride (III-N) heterostructure grown using molecular beam epitaxy (MBE) on a SiC substrate[31–33]. The TMN/III-N layers respectively combine a metallic NbN film (a superconductor with critical temperature $T_c < 16$ K)[31] with a piezoelectric GaN film. A schematic of the epi-HBAR is shown in Fig. 1a). Optimized epitaxy enables single-crystal films with low defect density (X-ray diffraction (XRD) rocking curve measurements show full width at half maximum values for the various layers are lower than 1000 arc-sec), and smooth interfaces and surfaces (roughness <0.9 nm). We experimentally show that the resulting epi-HBARs demonstrate extremely high mechanical $Q$ and phonon lifetimes. Measurements on the epi-HBARs exhibit $Q$ values as high as 13.6 million at a frequency ($f$) of 10 GHz (an $f \times Q$ product of $1.36 \times 10^{17}$ Hz) at 7.2 K, which are significantly better than other reported sputter-deposited HBARs with maximum $f \times Q$ products on the order of $\approx 4 \times 10^{15}$ Hz at cryogenic temperatures (T < 10 K)[5]. At room temperature, the epi-HBARs demonstrate a maximum $f \times Q$ product of $2.3 \times 10^{15}$ Hz, which is significantly higher than sputter-deposited HBARs operated at room temperature with reported $f \times Q$ products on the order of $\approx 2 \times 10^{14}$ Hz[18,21,28]. The GaN/AlN/NbN/SiC epi-HBAR demonstrated here has the potential for integrating acoustic phonons with superconducting qubits and other superconducting circuit elements utilizing the NbN layer[31], or with spin qubits in the SiC substrate[4]. Recent demonstrations by the authors have shown that the epi-HBAR can be monolithically integrated with conventional AlGaN/GaN high electron mobility transistors[34], combining the mode density of an epi-HBAR with on-chip electronic amplification and signal processing capabilities. Other attributes of the materials in the epi-HBAR (electromechanical, semiconducting, optoelectronic, superconducting etc.)[35,36] can be used for a number of multifunctional systems using the same heterostructure.

## Results

**Epitaxial HBARs with acoustic impedance matching.** The HBAR (conventionally sputter deposited or epitaxial) comprises of an electrode/piezoelectric/electrode transducer on a thicker low-loss substrate (Fig. 1a). A hypothetical 'un-attached' piezoelectric transducer (Supplementary Fig. 1) has a spectral response given by:

$$f_n \equiv \frac{\omega_n}{2\pi} = (2n+1)\left(\frac{v_r}{2t_r}\right) \quad ; n \in [0, 1, 2, \dots], \qquad (1)$$

where $\omega$, $v$, and $t$ denote radial frequency, longitudinal velocity, and thickness, and the subscript $r$ indicates the piezoelectric material. When attached to a substrate (denoted by the subscript $s$) with finite thickness, the piezoelectric transducer launches acoustic energy into the substrate, which acts as a phononic Fabry–Perot cavity. The cavity confines all $m$ phonon modes with

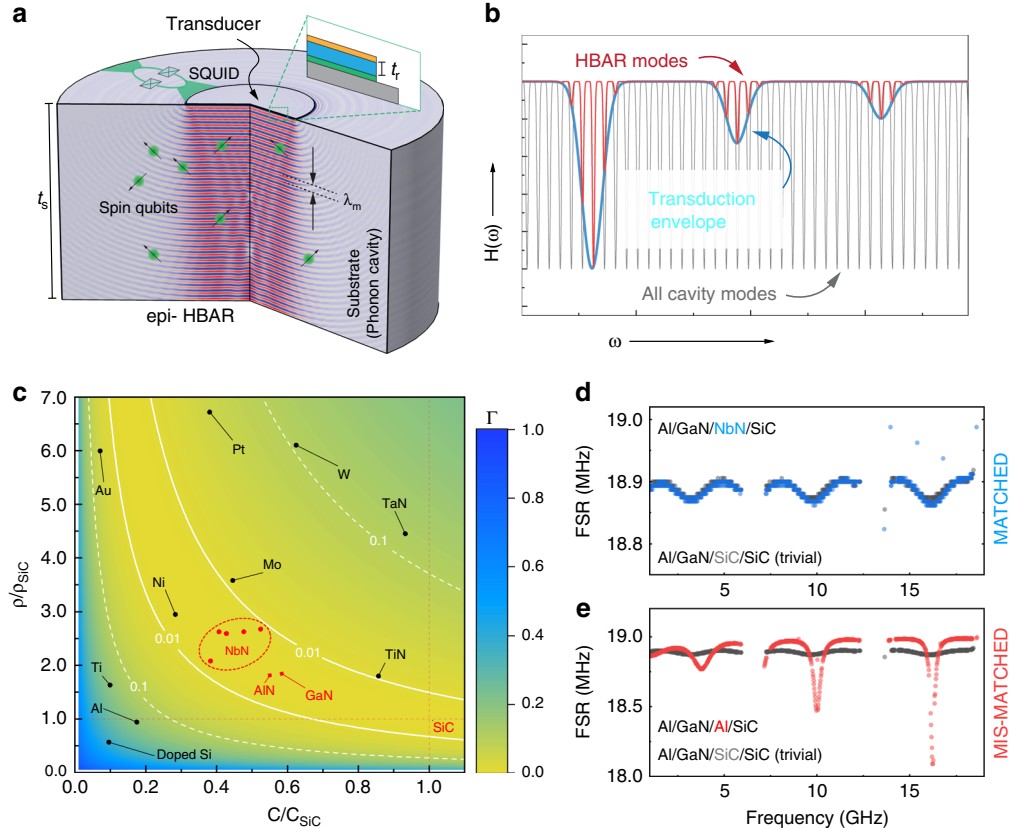

**Fig. 1 The epi-HBAR as a versatile and efficient multimode phonon source for quantum acoustodynamic systems. a** A depiction of phonon cavity modes of the epi-HBAR generated by the thin epitaxial piezoelectric transducer and confined in the substrate. Phonon–qubit coupling can be possible with planar/ vertical NbN-based superconducting qubits, or with spin qubits in the SiC substrate. **b** The net spectral response (red) of an epi-HBAR is the superposition of the transduction envelope (blue) of the electrode/piezoelectric/electrode transducer, and the cavity modes (gray) of the substrate. **c** The parameter space of the Fresnel acoustic power reflection $\Gamma$ as a function of stiffness and mass density of the bottom electrode, normalized to the values for the SiC substrate. All epitaxial materials in the epi-HBAR, especially the critical NbN bottom electrode, are acoustically impedance matched to the substrate (that is the power reflection coefficient $\Gamma < 0.01$), resulting in efficient acoustic power injection $(1 - \Gamma)$ into the phonon cavity. Calculated FSR spectra for (**d**) NbN and (**e**) Al bottom electrodes compared with the trivial solution with a SiC bottom electrode (perfectly matched acoustic impedance) demonstrates the difference between impedance matched and mismatched conditions. The FSR distribution for the calculated Al/GaN/NbN/SiC epi-HBAR agrees closely with the experimental data from Fig. 2. Note that (**d**), (**e**) do not have the same Y-axis scaling.

wavelengths $\lambda_m$ that meet the condition:

$$f_m \equiv \frac{\omega_m}{2\pi} = \left(\frac{v_s}{\lambda_m}\right) = m \times \left(\frac{v_s}{2t_s}\right). \quad (2)$$

The spectral response of the HBAR is the transfer function of the piezoelectric transducer loaded by the thick substrate, resulting in a comb-like phonon spectrum (Fig. 1b). The free spectral range (FSR) between successive HBAR modes is an inverse function of the substrate thickness. The interface between the bottom electrode (BE) and the substrate is crucial for efficient power transfer from the phonon source (transducer) to the phonon cavity (substrate). For longitudinal acoustic phonons generated in the transducer and normally incident at this interface, a low value for the Fresnel power reflection coefficient $(\Gamma \to 0)$ is possible if the characteristic acoustic impedances $(Z_{ac} = \sqrt{C\rho})$ of the BE and substrate are almost identical, where $C$ and $\rho$ are the longitudinal stiffness and mass density of the material respectively. The parameter space for $\Gamma$ (relative to SiC) indicates that the use of a dense and stiff metal like NbN yields exceedingly low reflection $(\Gamma < 0.01)$ back into the transducer, and consequently efficient power injection into the SiC substrate (Fig. 1c). The use of other common electrode metals such as Al, Pt, Mo, and Ti result in higher reflection $(\Gamma > 0.1)$. Figure 1d, e presents a numerical comparison of the FSR distribution of the

Al/GaN/NbN/SiC epi-HBAR with $\Gamma < 0.01$ and a hypothetical Al/ GaN/Al/SiC HBAR with $\Gamma > 0.1$. Both GaN and AlN are also acoustically impedance matched to SiC, with $\Gamma < 0.01$. The sinusoidal variation in FSR seen in the NbN case (Fig. 1d) is a signature of acoustic impedance matching with low reflection $(\Gamma < 0.01)$, while the sharper FSR variation seen in Fig. 1e is indicative of a mismatch[37,38] (Supplementary Fig. 2). For the epi-HBARs measured in this work, the use of acoustically matched GaN/AlN/NbN/SiC interfaces results in power injection efficiency $(1 - \Gamma) > 0.99$, which is critical in applications such as acousto-optical modulators or acoustic spin pumping, where the stress amplitude in the substrate is important[10–13,26].

**Spectral measurements.** The epi-HBAR microwave reflection spectrum $(S_{11}(\omega))$ from 1 to 17 GHz (Fig. 2) confirms the existence of many individual phonon modes $(m \in [52, 897])$ with exquisitely narrow linewidths. Unless otherwise specified, the data were acquired at 7.2 K (Methods). The epi-HBAR phonon modes form a regular phononic comb spectrum (Fig. 2a) enveloped by three odd modes of the transducer, $f_1 = 3.1$ GHz, $f_3 = 9.3$ GHz, and $f_4 = 15.5$ GHz (Supplementary Fig. 1). Figure 2b shows a magnified range of the phononic spectrum, clearly showing the sharp periodic epi-HBAR phonon modes separated by an FSR of ~18.95 MHz. Figure 2c shows the impedance $Re|Z_{11}(\omega)|(\Omega)$

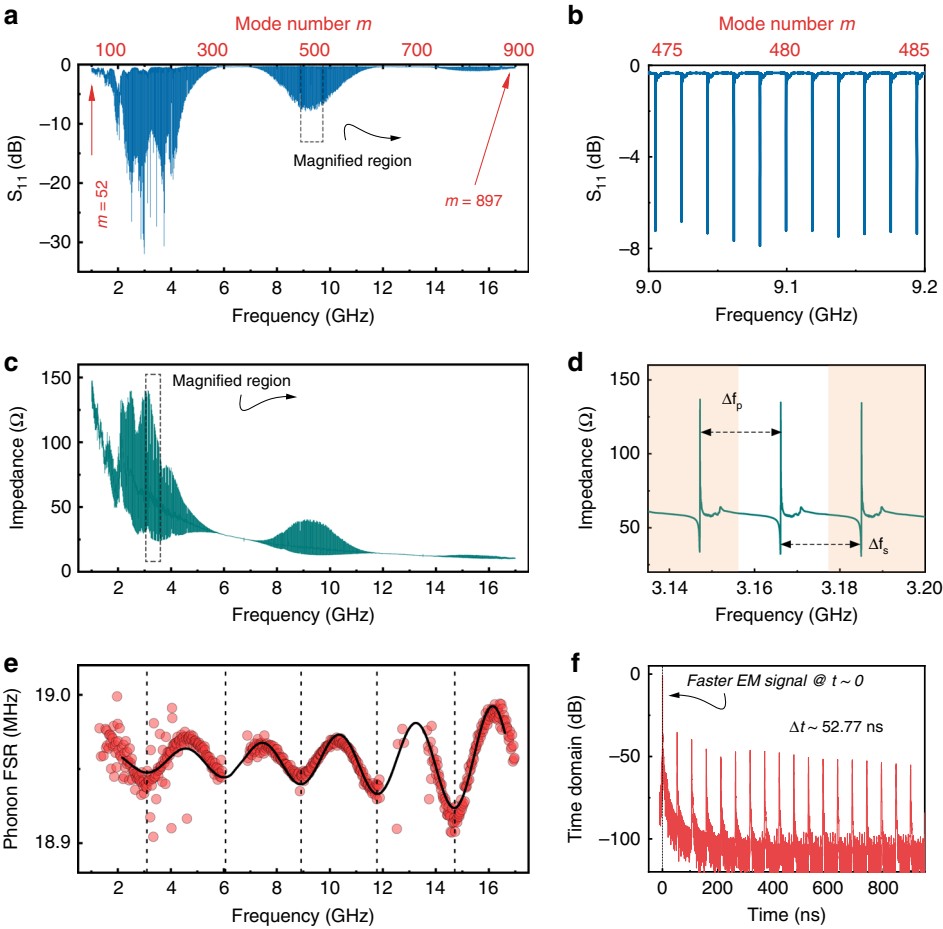

**Fig. 2 Microwave measurements of the epi-HBAR at 7.2 K. a** Microwave reflection spectrum ($S_{11}(\omega)$) for the epi-HBAR from 1 to 17 GHz (encompassing $m = 52$ to $m = 897$). The $S_{11}(\omega)$ spectrum demonstrates the superposition of the transduction envelope and the cavity modes. The transduction envelope encompasses three odd-numbered modes of the Al/GaN/NbN transducer. At low frequencies (<2.5 GHz), the presence of unwanted SAW or shear modes distorts the spectral response slightly. **b** Magnified range of $S_{11}(\omega)$ clearly shows periodic, sharp epi-HBAR phonon modes separated by an average FSR of ~18.95 MHz. **c** The electromechanical impedance $|Z_{11}(\omega)|$ has a global hyperbolic response (from the static electrical capacitance), overlaid by the impedance of the epi-HBAR phonon modes. **d** A magnified section of $|Z_{11}(\omega)|$ illustrates the series (parallel) resonance $f_s(f_p)$ corresponding to the minimum (maximum) impedance. The mode separation between adjacent resonances $\Delta f_s$ (or $\Delta f_p$) is the free spectral range (FSR) of the epi-HBAR. **e** Due to the composite nature of the epi-HBAR, the FSR is not constant across the spectrum, but good acoustic impedance matching ensures a smooth sinusoidal dependence with good power transfer, as discussed in Fig. 1c–e. **f** A fast Fourier transform of $S_{11}(\omega)$ yields the time-domain response of the epi-HBAR, which is comprised of an electromagnetic reflection signal at $t \to 0$, followed by a pulse-train of phonons separated by a mode delay of $\Delta t = (\Delta f)^{-1} \approx 52.77\,ns$.

across the spectral range. A control sample with an AlGaN/GaN/SiC heterostructure confirms that no epi-HBAR phonon modes are observed in the absence of the NbN layer, highlighting the critical importance of a BE (Supplementary Table 1 and Supplementary Figs. 1, 3, 4). The distortion for $f < 2.5$ GHz could be a result of the transduction of unintended SAW modes or other unintended shear waves actuated in the heterostructure. Figure 2d shows a magnified impedance plot, with each individual phonon mode characterized by a series and parallel resonances ($f_s$, $f_p$). The FSR of the epi-HBAR is given by $\Delta f = f_{m+1} - f_m$, with a corresponding time-domain pulse response spaced at $\Delta t = (\Delta f)^{-1}$ (Fig. 2f). The FSR distribution for the epi-HBAR (Fig. 2e) is found to be proportional to $\sin(\gamma)$, for $\gamma = (\omega t_r/v_r)$, proving that the epi-HBAR has excellent acoustic impedance matching, as expected from Fig. 1c–e.

**Quality factor and phonon relaxation time.** The quality factors extracted from the measured data (Methods) are shown in Fig. 3a, with $Q \to 10^7$ at frequencies above 8 GHz. Since $Q$ is often a

strong function of frequency, the product of frequency and quality factor ($f \times Q$) is used as a figure of merit (Fig. 3b). As expected, the $f \times Q$ trend follows the Landau–Rumer regime of phonon scattering (Supplementary Table 2, Supplementary Fig. 5), where $(f \times Q) \propto f$ at a fixed temperature, and $f \times Q \to 10^{17}$ Hz. The $f \times Q$ product is inversely proportional to the total attenuation $\alpha$, which is calculated to be $\alpha < 2$ dB.m$^{-1}$ or equivalently, $\alpha < 10^{-5}$ dB $\lambda_m^{-1}$ (Methods). The phonon relaxation time for each mode is given by $\tau = 2Q/\omega_m$, and $\tau \to 500\,\mu s$ (Fig. 3c)[23]. Each epi-HBAR phonon mode can also be represented by the Butterworth-van Dyke (BVD) electromechanical equivalent circuit model (Fig. 3d). Three representative phonon modes ($m = 164$, $m = 476$, $m = 529$) along with BVD model fits are shown in Fig. 3e–g, and extracted $Q_{BVD}$ values agree well with measured $Q$ (Methods). Corresponding admittance $|Y_{11}(\omega)|$ and impedance $|Z_{11}(\omega)|$ spectra, along with BVD model fits for these modes are provided in Supplementary Fig. 6. The measured values of $f \times Q$ and $\tau$ for representative modes ($m = (171, 239, 476, 529)$) of the epi-HBAR, at various temperatures

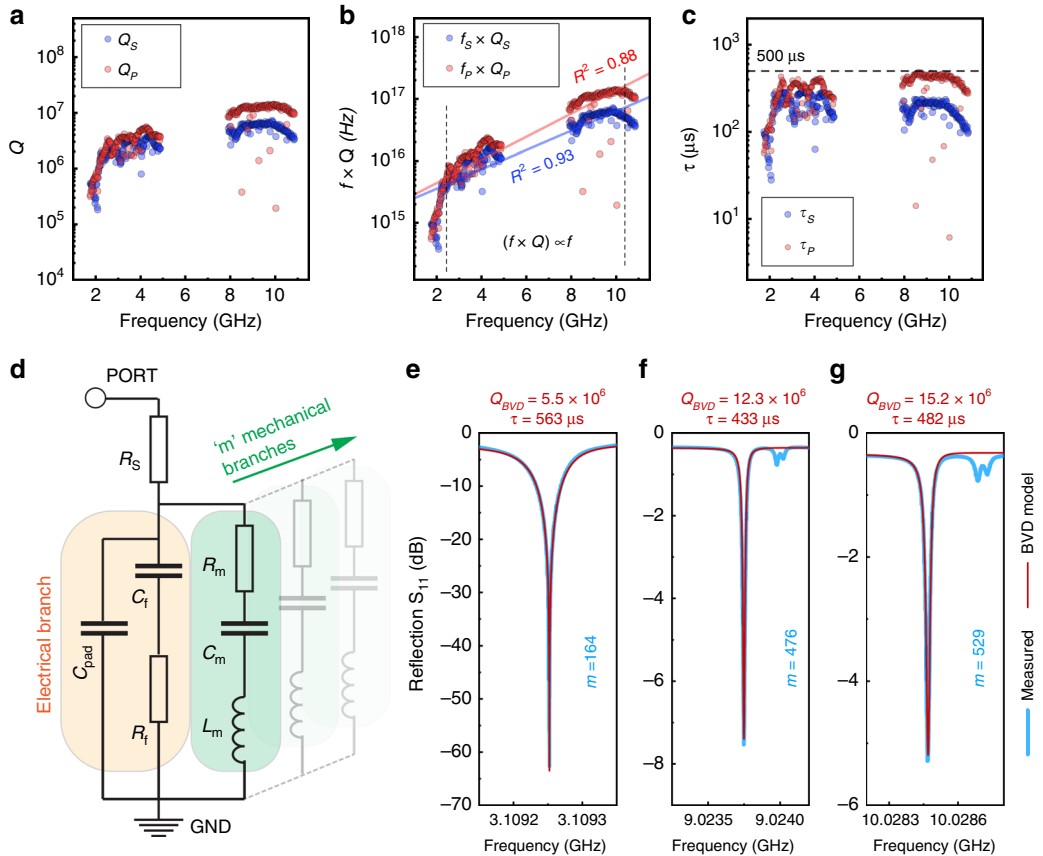

**Fig. 3 High mechanical quality factors and phonon lifetimes for the epi-HBAR. a** Measured quality factors (for phonon modes that exhibit at least a half-power bandwidth) for the epi-HBAR are greater than 10 million (at frequencies greater than 10 GHz) at 7.2 K, some of the best values measured to date for BAW resonators. **b** The $f \times Q$ products show that $(f \times Q) \propto f$, characteristic of the Landau–Rumer regime for anharmonic phonon scattering. The dashed lines denote linear fits, with goodness of fit ($R^2$) values of 0.93 and 0.88 for series and parallel modes respectively. **c** The phonon relaxation times ($\tau$) for the measured epi-HBAR is as high as 500 μs. The phonon relaxation time effectively sets the upper bound for the time that a quantum state can be stored or manipulated in a QAD system. **d** A Butterworth-van Dyke (BVD) equivalent circuit is used to model multimode mechanical resonators. Each mechanical branch is modeled as a virtual resonant circuit that represents a single phonon mode; the electrical branch models all electromagnetic losses and feedthrough associated with the epi-HBAR and the coplanar waveguide. Panels (**e**–**g**) show measured $S_{11}(\omega)$ for three epi-HBAR modes ($m = 164$, $m = 476$, and $m = 529$) (blue) along with corresponding BVD model fits (red). Measured and modeled quality factor and relaxation time (Methods) for each mode validates the high $f \times Q$ and $\tau$, demonstrating $f \times Q > 10^{17}$ Hz and $\tau > 500$ μs, respectively, for microwave phonons at 7.2 K in Al/GaN/NbN/SiC epi-HBARs.

(7.2 K – room temperature), are compared with a variety of sputter-deposited HBARs and overtone BAW resonators reported in the literature over temperatures ranging from 0.02 K to room temperature (Fig. 4a, b) (Methods). The only devices that are comparable or better than the epi-HBARs are the massive quartz overtone BAW resonators (1 − 5 mm thick, 13 mm diameter) ground into confocal shapes for phonon focusing, and measured between 0.02 and 10 K[6,7,17,25]. Many of these macroscale BAW resonators use fully optomechanical transduction[6,7,25], eliminating the need for conducting electrodes and the resultant material interfaces, and consequently demonstrate high phonon relaxation times. While these massive BAW devices have the advantage of acoustic focusing and relative ease of manufacture, they have a large footprint and are not easy to integrate into solid-state systems. The use of epi-HBARs is an attractive alternative that provides nearly the same long phonon relaxation lifetimes in a much smaller mode volume, while providing for better integration and functionality. To eliminate the effect of the low-loss substrate in this survey, the epi-HBARs are explicitly compared with polycrystalline HBARs sputter-deposited on SiC (Fig. 4c), clearly demonstrating that the epi-HBARs demonstrate longer phonon lifetimes than sputter-deposited HBARs. Comparing only

HBARs sputter-deposited on a variety of low-loss substrates, we see no clear trend indicating that the choice of substrate has a defining influence on lifetime (Fig. 4d). The performance of the epi-HBARs even at room temperature is better than most sputter-deposited HBARs at cryogenic temperatures. Room temperature performance of the epi-HBARs and a subset of the comparison of $f \times Q$ and $\tau$ for specific temperature ranges of interest is given in Supplementary Figs. 7 and 8. Material characterization was carried out on the epi-HBAR using electron microscopy, atomic force microscopy (AFM), and XRD, revealing an epitaxial heterostructure with high crystallinity, low defect density, clean interfaces and consistent texture, and atomically smooth surfaces with sub-nanometer rms roughness (Methods, Supplementary Figs. 9–11). These analyses strongly confirm the high quality of the epi-HBAR heterostructure and validate our hypothesis that the epitaxially grown, acoustically matched epi-HBARs provide superior performance.

## Discussion

A recently proposed QAD computing architecture envisions a single (or a few) qubit(s) coupled to multiple nanomechanical

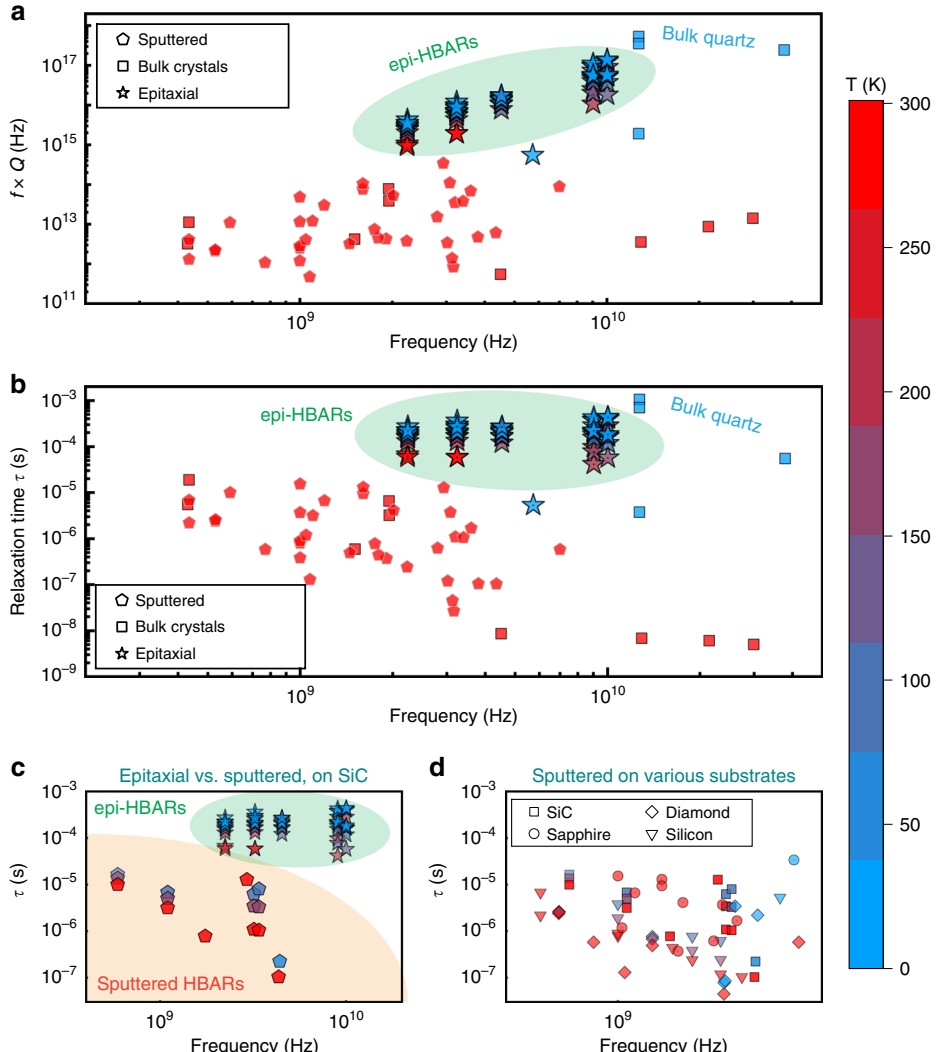

**Fig. 4 The case for epi-HBARs.** A comparison of (**a**) the $f \times Q$ product and (**b**) phonon relaxation time ($\tau$) of selected epi-HBAR phonon modes (stars) from this work, with published values for overtone mode solid BAW resonators (thick, millimeter scale single-crystal bulk devices) (squares) and sputter-deposited HBARs (thin film polycrystalline transducers on low-loss substrates) (pentagons). The measurement temperature for each data point is indicated by color. Only massive plano-convex quartz bulk resonators[6,7] using optomechanical transduction, and measured at cryogenic temperatures, demonstrate phonon relaxation times higher than that of the epi-HBARs, which exhibit $\tau > 500$ µs for microwave frequencies. **c** A comparison of epi-HBARs and sputtered HBARs, both with SiC as the substrate, validates that even with low-loss substrates the quality of the transducer heterostructure is critical to achieving high phonon lifetimes. **d** A study of only sputter-deposited HBARs on various low-loss substrates does not clearly indicate the superiority of any particular substrate material.

resonators, with the goals of reducing technological complexity and computational bottlenecks, and improving computational efficiency[9]. From a practical perspective, such a QAD architecture should have a compact form-factor with electronic drive/readout interfaces. A critical requirement is an ensemble of electrically pumped phonon sources with high coherence, low crosstalk (that is, sharp linewidths and clear mode separation), and a spectral range that allows for easy coupling with qubits. Unlike conventional low-frequency flexural/SAW/BAW resonators, the epi-HBAR results in a large number of sharp periodically spaced phonon modes. The epi-HBAR can be thought of as a geometrically compact ensemble of $m$ spectrally periodic electrically pumped phonon sources, where each $m^{th}$ mode can be individually addressed by a qubit at the corresponding frequency. While an ensemble of individual resonators might suffer from variations across the ensemble, the modes of an epi-HBAR have a strict and predictable internal relationship. Recent QAD experiments have demonstrated up to $\tau \approx 1.5$ s for a single-mode

engineered silicon cavity using a laser pump-probe experiment ($Q > 10^{10}$ at 5 GHz and $T \sim 10$ mK)[39]. The electrically transduced epi-HBAR data here were acquired at 7.2 K. Depending on the dominant damping regime (Akhieser or Landau–Rumer), $f \times Q$ is expected to scale proportional to $\sim T^{-1}$ or $\sim T^{-4}$ at the millikelvin temperatures required to operate most quantum systems. We expect scattering losses to continue influencing epi-HBARs to some extent, making further $T^{-4}$ improvement unrealistically optimistic. However, even a conservative improvement by a factor of $T^{-1}$ would result in $Q \to 10^{10}$, $f \times Q \to 10^{21}$ Hz and microwave phonons with $\tau \to 10$ s if epi-HBARs are operated at millikelvin temperature regimes. Other attributes of the materials in the epi-HBARs (superconducting, spintronic, semiconducting, optoelectronic, etc.) can be utilized in the future for making a high-performance multifunctional quantum system. The use of the epi-HBAR in such a QAD architecture can lead to a compact and robust hybrid quantum system, easy to fabricate and interface, capable of quantum information storage and processing over

time-scales large enough for practical utility. In addition to QAD systems, epi-HBARs with demonstrated high performance at room temperature ($f \times Q > 10^{15}$ Hz, ~10× higher than conventional HBARs) can be used as subcomponents of conventional RF signal processing systems, primarily mechanical filters and oscillators operating at $f > 5$ GHz. The high $f \times Q$ can enable sharp, multimode channel select filters integrated with GaN electronics[34,40–42] and lead to stable low-jitter mechanical oscillators locked to a chip-scale atomic clock[27].

## Methods

**Epitaxy and device fabrication.** The material system we use in this work is a Group III-Nitride (III-N) piezoelectric semiconductor grown on a TMN film. The AlGaN/GaN/NbN/SiC heterostructure reported here has been used to demonstrate a two-dimensional electron gas (2DEG) with a 2D carrier density of $6.9 \times 10^{12}$ cm$^{-2}$ and a mobility of 1000 cm$^2$V$^{-1}$s$^{-1}$ at 300 K[34]. The epitaxial layers were grown by RF plasma MBE using a system and procedures described earlier[32,33,43]. In this sample, the polarity of the GaN was converted to Ga-polar using a 45-nm-thick low-temperature AlN nucleation layer grown directly on the 50-nm-thick NbN BE layer. Next, 1.2 μm of Ga-polar GaN was grown, followed by a 25-nm-thick Al$_{0.29}$Ga$_{0.73}$N barrier layer[31]. The control sample has a nearly identical heterostructure and properties, with a crucial exception: the absence of the NbN BE. While the epitaxial approach potentially reduces the pool of materials available for the various layers in an HBAR, recent progress by the authors and others have shown that a large number of high-performance candidate piezoelectric materials are available[31,32,43–45]. Within reason, we can potentially substitute an appropriate combination of III-N materials (AlN, InN, GaN, AlGaN, ScAlN), with any combination of stiff refractory TMN (TiN, NbN, TaN, WN), grown on 4H-SiC, 6H-SiC, bulk-GaN, sapphire, or diamond. Many of these viable combinations have been successfully fabricated and are being actively investigated by the authors for various other applications[31,32,43–49]. After MBE growth, the epi-HBAR is fabricated in two simple steps. The AlGaN barrier layer was etched away using a Cl$_2$/BCl$_3$ plasma etch in areas where HBAR devices were fabricated. The epi-HBAR reported in this work is a simple square electrode of size $100 \times 100$ μm. The top electrode of the epi-HBAR is a 10 nm/50 nm Cr/Al layer deposited using electron beam evaporation and optical lithography. The top electrode is the only component in the epi-HBAR that is not currently epitaxially grown, although future implementations can easily produce a TMN/III-N/TMN transducer if needed. For on-chip probing, a coplanar waveguide (CPW) optimized for 50 Ω input impedance with a thick Au layer (200 nm) is also deposited using electron beam deposition and liftoff.

**Acoustic impedance matching to SiC.** For longitudinal acoustic phonons generated by the transducer and normally incident at the BE/substrate interface, the Fresnel power reflection coefficient is given by $\Gamma = |(Z_b - Z_{sub})/(Z_b + Z_{sub})|^2$. When $Z_b \rightarrow Z_{sub}$, the reflection coefficient $\Gamma \rightarrow 0$, and all the power generated in the piezoelectric transducer is transmitted to the substrate. This is the basis for calculating the parameter space shown in Fig. 1c. The acoustic impedance model for a composite overtone resonator or HBAR is described in detail in well-established models by Zhang, Cheeke, and Hickernell[37,38,50]. In essence, it uses the mechanical properties of four layers (top electrode, piezoelectric layer, BE, and substrate) in a matrix formulation in order to solve for the input impedance of the HBAR. We use this numerical model to solve for the input electromechanical impedance $Z_{11}(\omega)$ of an epi-HBAR given by Al/GaN/AlN/(BE)/SiC, where the BE is any metal that is commonly used as an electrode material in nanoresonators. Film thicknesses for each layer are fixed as specified in Supplementary Table 1. We calculate the FSR distribution for each BE material, including for an entirely hypothetical situation where the BE has the exact mechanical properties of SiC. For further ease of comparison, the FSR spectrum for each case is normalized to that of the hypothetical SiC BE. As expected from the impedance matching parameter space in Fig. 1c, Ni and NbN BE are acoustically well matched to the SiC substrate, while Al, W, and Pt present some of the worst mismatches (Supplementary Fig. 2).

**Measurement and simulations of the epi-HBAR.** The fabricated epi-HBAR is mounted in a helium-flow cryostat probe station (Lakeshore Cryotronics). The cryostat is pumped down to a pressure below $10^{-6}$ Torr, cooled down to the lowest achievable base temperature (7.2 K), and allowed to stabilize. A single microwave probe with a ground-signal-ground configuration is used to contact the thick CPW input and to apply the electrical input to the epi-HBAR. Electrical input and readout is provided by a Keysight vector network analyzer (VNA) with 50 Ω port termination. The effect of the microwave probes and cables are removed by using a calibration standard and a short-open-load (SOL) calibration sequence. Note that all spectral measurements are presented as is, with no de-embedding or removal of parasitic elements. The electrical resistances can be further reduced by optimizing the thickness/resistivity of the electrodes, and potentially eliminated by operating in the superconducting regime. In order to achieve sufficient spectral resolution the wide spectral measurements shown here are composites of multiple sequential scans acquired using automated control of the VNA using LabVIEW. The spans of

each individual scan were between 250 kHz and 250 MHz, with each scan containing 32001 data-points (limited by the VNA) at an acquisition speed of 1 kHz. Microwave measurements acquired at room temperature are presented in (Supplementary Fig. 7). All results presented here are for a microwave input power of −15 dBm (31.62 μW) and no significant variation or non-linearity was observed in the results up to input powers of up to 0 dBm (1 mW). We simulate the spectral response of a hypothetical Al/GaN/NbN transducer that is unattached to any substrate (Supplementary Fig. 1). This is effectively a film bulk acoustic resonator with free-free mechanical boundary conditions. This is in order to verify that the spectral responses for the transducer modes $f_{r,i}$ for the appropriate film thicknesses are observed at approximately the correct frequencies. Finite element analysis is carried out using COMSOL Multiphysics. The control sample without the NbN electrode (from Supplementary Fig. 3 and Supplementary Table 1) is used to compare with the Al/GaN/AlN/NbN/SiC epi-HBAR. In addition, interdigitated SAW cavity devices with varying wavelengths are fabricated on the control sample. The comparison serves two purposes: (a) it verifies the presence and approximate frequencies of unwanted modes (potentially SAW modes) in the $f < 2.5$ GHz range in the epi-HBAR, and (b) it demonstrates that without the NbN BE, HBAR modes are not actuated at all, leading to a clean SAW response. The various SAW modes in Supplementary Fig. 4b correspond to Rayleigh ($R_\Lambda$), and Sezawa ($S_\Lambda$) modes, where the SAW wavelength is given by $\Lambda \in [3, 4, 5] \mu m$. Finite element simulations of the epi-HBAR (Supplementary Fig. 4c) and the control sample (Supplementary Fig. 4d), respectively, verify that the presence or absence of the BE makes a significant change in the acoustic behavior of the device.

**Mechanical quality factors of the epi-HBAR: extraction and modeling.** The series and parallel quality factors of each phonon mode are calculated from the impedance spectrum as $Q_{s,p} = \left(\frac{f}{2}\right)\left|\frac{\delta\phi_z}{\delta f}\right|_{s,p}$, where $\phi_z$ is the phase of the impedance. In order to avoid inaccurate estimates of $Q$ for low amplitude or noisy data, we only report $Q$ for phonon modes with a valid half-power bandwidth. This specifically affects measured data in the 14–17 GHz range, where we do not report $Q$ values even though we see clear evidence for the existence of periodic phonon modes. Given that the HBAR envelope from 14 to 17 GHz corresponds to $f_{r,5}$, the fifth harmonic mode of the piezoelectric transducer, we expect, and observe a low signal strength for all phonon modes in this range. We use a one-dimensional electromechanical equivalent model, the BVD model, to model and fit the epi-HBAR. The admittance and impedance values for three representative modes ($m = (164, 476,$ and 529)) of the epi-HBAR are shown in Supplementary Fig. 6, along with the corresponding BVD model fits. The mechanical branch of the BVD model for each mode represents the mechanical resonance condition $\omega_m = 1/\sqrt{L_m C_m}$, with loss represented by the resistance $R_m$. The electrical branch includes all electrical resistances and feedthrough signals associated with the structure of the HBAR and the microwave contact pads. The unloaded mechanical quality factor derived from the BVD model is given by $Q_{BVD} = \omega_m L_m / R_m = 1/\omega_m C_m R_m$. While the measured $Q_{s,p}$ includes any dissipative elements in the electronic readout interface, the 'unloaded' quality factor, $Q_{BVD}$ models only mechanical losses inherent to the body of the phonon cavity. Thus, it is expected that $Q_{BVD} > Q_{s,p}$. As expected, the values of $Q_{BVD}$ are slightly higher than, but are close to the corresponding measured $Q_{s,p}$. The high values of both measured $Q_{s,p}$ and $Q_{BVD}$ present strong evidence that a number of phonon modes at microwave frequencies have extremely low levels of total phonon loss, corresponding to $Q > 10^7$. Similarly, the phonon relaxation time extracted from the BVD model agrees with calculated values, indicating $\tau > 500$ μs. The mechanical quality factor of a resonator is inversely related to the total effective loss; that is, $Q = \omega/(2\alpha v)$ where $v$ denotes the phase velocity for the mode in question, and $\alpha$ is the attenuation per unit length.

**Anharmonic phonon loss limits.** For phononic devices, loss arising from the anharmonicity of the crystal lattice is a fundamental energy loss mechanism. Anharmonic losses are caused by the interaction between acoustic waves (or coherent acoustic phonons) and the thermal phonon distribution in the material. In ideal crystal lattices, anharmonic phonon scattering and $f \times Q$ are described by two distinct regimes across frequency and temperature: the low-frequency Akhieser regime ($(f \times Q) \propto T^{-1}$) and the Landau–Rumer regime ($(f \times Q) \propto fT^{-4}$). It is important to note that these anharmonic losses are separate from intrinsic defect mediated losses, and will occur even in ideal defect-free crystals at temperatures greater than absolute zero. As such, anharmonic losses pose the ultimate theoretical energy loss limits for BAW devices. For the spectral region $\omega\tau_{th} < 1$, where $\tau_{th}$ is the thermal phonon lifetime, the loss can be described by classical acoustic waves losing energy to a bath of thermal phonons with much shorter wavelengths than the acoustic wave. This model was first developed by Akhieser[23,51], and is popularly referred to as the Akhieser loss. For the spectral region given by $\omega\tau_{th} > 1$ (the Landau–Rumer regime), the acoustic phonons generated by the device are damped by interaction with individual thermal phonons with comparable wavelengths. Analytical models along with material properties of SiC used to calculate the maximum $f \times Q$ for the SiC substrate as a function of temperature and frequency (Supplementary Table 2, Supplementary Fig. 5). We emphasize that these values are theoretical estimates for perfect 4H-SiC; $f \times Q$ values for practical devices, even epi-HBARs on SiC, are expected to be lower.

**Total phonon attenuation**. In practice, the total phonon attenuation in a mechanical resonator is attributable to the combined effect of a number of extrinsic loss mechanisms, including gas damping, thermoelastic damping, and anchor loss, as well as intrinsic loss mechanisms such as phonon–electron scattering, anharmonic phonon scattering, and phonon scattering as a result of surface/interface roughness, film defects, and grain boundaries[23,52–59]. For resonators operating in vacuum, we can eliminate gas damping. For high-frequency longitudinal phonons in HBARs, thermoelastic damping is similarly negligible. For the epi-HBAR, the bulk of the piezoelectric transducer is unintentionally doped GaN and thus the phonon–electron scattering is not expected to be the dominant loss mechanism[52]. This assumption is reinforced by the fact that the measured $f \times Q$ for the Al/GaN/AlN/NbN/SiC epi-HBARs is consistently higher than values for HBARs with undoped ZnO or AlN on SiC. If the boundary conditions of the phonon cavity were ideal, anchor loss would be eliminated. The anharmonic phonon scattering in the low-loss substrate is the ultimate theoretical limit of phonon loss in the epi-HBAR, but this theoretical limit is generally overshadowed by losses caused by the transducer microstructure (surface/interface roughness, film defects, and grain boundaries), which are thus the major impediment to achieving peak performance. The granularity or texture of the thin film has a large impact on the total attenuation of the longitudinal HBAR modes. In polycrystalline films with misaligned crystalline axes, each individual grain with a local tilt produces a local shear wave that reduces the energy of the HBAR mode. Experimental evidence acquired using sputter-deposited AlN indicates that phonon attenuation scales proportionally to oxygen content and defect density, thickness and phase variability, and stress gradients[57,58]. In contrast, the use of ultrahigh vacuum techniques such as MBE help minimize impurities like oxygen in the epi-HBAR microstructure. Attenuation caused by surface roughness of the thin films scales as $\alpha_{SR} \propto (z^2/\lambda_m^2)$, where $z$ is the RMS surface roughness[54,55], making the surface and interface quality of the films more critical at higher frequencies.

**Comparison with other HBAR and BAW resonators**. Apart from the GaN/NbN/SiC epi-HBAR data in this work, our study includes data reported in the literature from solid BAW overtone resonators fabricated from single-crystal bulk quartz, LiNbO₃, and Si[6,7,17,24,25,42,60–62], as well as sputter-deposited HBARs on diamond[13,16,20,26,30], fused silica[14], quartz[14], sapphire[5,14,15,21,27,28,63], silicon[14,15,26,29], and SiC[16,18,19,26] substrates. The sputter-deposited HBARs use polycrystalline piezoelectric AlN and ZnO thin films deposited on Al, Pt, Mo, and p⁺ doped Si BE. One other reported epitaxial HBAR uses hybrid vapor phase epitaxy (HVPE) GaN grown on doped Si[10], which also exhibits an $f \times Q$ value higher than most sputter-deposited HBARs (but not as high as the epi-HBARs on SiC from this work). Apart from the comparison between epi-HBARs and HBARs on SiC, we also investigate the effects of various substrates, piezoelectric films, and top electrode materials. No significant trend is seen to indicate that the choice of a specific substrate material provides significantly superior performance. To emphasize the higher performance of the epi-HBARs from this work with respect to other reported HBARs and BAW resonators, Supplementary Fig. 8a, b compares only data for a cryogenic temperature range ($T < 10$ K), while Supplementary Fig. 8c, d compares only data reported at room temperature.

**Material characterization**. The MBE grown AlGaN/GaN/AlN/NbN/SiC heterostructure used in this work is characterized using a number of techniques. Samples for transmission electron microscopy (TEM) analysis were prepared via an in-situ liftout procedure on a FEI Helios G3 focused ion beam. The lift-out was performed using a 30 kV ion beam, and the sample was thinned to electron transparency using successive 5 and 2 kV ions to reduce implantation damage in the final specimen. TEM and HRTEM imaging was done using a JEOL 2200FS equipped with an in-column Omega filter and ultrahigh resolution pole piece. Selected area electron diffraction (SAED) patterns were obtained with a 100-nm diameter aperture and energy-filtered with a 20 eV slit inserted about the zero-loss peak. The choice of a set of near-lattice matched materials[32] and optimized MBE growth leads to high crystal quality and low roughness in all epitaxial layers as evidenced by TEM analysis (Supplementary Fig. 9). The GaN layer exhibits a dislocation density typical for GaN grown on SiC, and the micro-rotation between grains is typically less than 1°. The images show clean interfaces and consistent texture between consecutive epitaxial thin films as indicated by SAED images acquired at the various growth interfaces. As evidenced by the HRTEM, the epi-HBARs benefit from almost atomically smooth interfaces. Of particular interest is the clean and smooth NbN/SiC interface, which forms the logical boundary between the piezoelectrically active transducer and the passive substrate cavity, and is critical for efficient power transfer. XRD data were acquired using a Rigaku system that employs a rotating Cu anode to produce Cu-K$_\alpha$ radiation (Supplementary Fig. 10). The XRD studies of the epi-HBAR confirm the content of the epitaxially grown AlGaN/GaN/AlN/NbN/SiC heterostructure. Rocking curve analysis of each individual material shows the relative quality of the films. As expected, the SiC substrate is of excellent quality, with a full-width half maximum (FWHM) of the (0004) rocking curve of only 42 arc-sec. Successively grown epitaxial films have only slightly worse FWHM, 715 arc-sec and 997 arc-sec for the (111) NbN and (0002) GaN rocking curves, respectively, which confirms the high crystal quality of the layers. The surface morphology was measured using a Bruker Dimension FastScan AFM in the tapping mode. AFM studies of the first epitaxial layer (NbN) and the final epitaxial layer (AlGaN) show

that the films consistently demonstrate RMS surface roughness $z < 0.9$ nm (Supplementary Fig. 11). Since the entire heterostructure is grown in one continuous growth run without removing the sample from the MBE chamber, AFM data on NbN are acquired on a representative NbN/SiC sample with the same NbN thickness (50 nm) grown under the same conditions. This finding indicates that there is no significant degradation of the surface quality over sequential epitaxial growth.

## Data availability

Supplementary information is available for this paper. The data that support the findings of this study are available from the authors on reasonable request.

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

## Acknowledgements
This research was funded by the Office of Naval Research and was conducted while V.J. G. and A.C.L held Associateships from the National Research Council and the American Society for Engineering Education respectively. We would also like to thank Dr Laura Ruppalt (US Naval Research Laboratory) for access to the vacuum cryostat for low-temperature measurements.

## Author contributions
V.J.G. and B.P.D. designed and fabricated the epi-HBARs. D.S.K. grew and characterized the heterostructures. N.N. conducted AFM and XRD measurements and analysis. A.C.L. and R.M.S. performed TEM, HRTEM, and SAED analysis. V.J.G. carried out low-temperature measurements. V.J.G., B.P.D., and D.J.M. conducted other data analyses, and wrote and revised the manuscript with input from all other co-authors.

## Competing interests
The authors declare no competing interests.
