## [Peer Review File · Nature Communications]

Reviewers' comments:

Reviewer #1 (Remarks to the Author):

Abstract : the authors use several times vague terms like low, good, smooth, which to my own do not contribute positively to the paper quality. Although often read even in high quality papers, i would suggest to avoid using such terms without a quantification just as the authors do for phonon lifetime. The sentence "to achieve strong coupling and reduce..." is the type of example i would take care to avoid - strong coupling does not mean a thing, as it refers to nothing. Quartz is less coupled than LiTaO₃ but the coupling can be considered strong for given cuts compared to other. Also it may useful to to write "to reduce" or "reduced" to the best of my English basis.

"significantly reducing footprint, power consumption, and complexity.": compared to what ?

Page 2, top, HBAR have been studied for the mentioned application but i do not know an effective industrial application of such devices which remains marginal in the corresponding industry (although extremely exciting).

Again, i am extremely sensitive to the mention of "g=high" Q factors without precising the values and the operating frequency.

Classical BAW exhibit Q factors of about 2 millions at 5 MHz for instance whereas Q factors of HBARs may reach 40000 at 2 GHz using single crystals with direct electrical coupling, see for instance Pijolat & al publications (which is not mentioned in the reference list).

Better than low-loss substrate, the authors should mentioned substrates with minimal viscoelastic losses as i understand it is actually what is looked for.

The last paragraph of the introduction is an example of wording that does not contribute to science and reads like opinions as there is no quantification of Q×frequency and relaxation delay reference to which the authors results can be compared. It would be helpful to give some figures allowing for fixing the comparison, to evaluate the gain of Q×frequency product (never talk about Q factor without mentioning the operating frequency) and to objectively appreciate the author contribution.

on the same page (bottom) i read again a mention to a good acoustic matching, whereas a simple reference to "acoustic matching" is clear and sufficient : layers are matched together as reflection coefficient is lower than for instance 1%.

"exquisitely narrow linewidths" means with Q factors exceeding the usual values for acoustic based devices (and in parenthesis, give a Q.F product estimate/ $Q.F \gg 10^{14}$ which is a representative value published by Lakin in early 90's).

Q values of 10 millions are mentioned without mention to the frequency, it must be added: 10 GHz as the authors are focusing on the 3rd harmonics of the piezoelectric layer fundamental mode.

I stop commenting these flaws as there is a lot to edit all along the paper.

On fig.4, reference is done to Lithium Niobate. Some precision would be welcome when showing such a graph without clear referenceto previous work, which is in definitive difficult to interpret and to verify.

I want to mention that the results of the authors are remarkable and that they must present them at best, with a very care to avoid diffuse mentions and comparative adjectives without reference. Probably it would be of interest to mention the actual Q×f values they obtained at room temperature, as from my opinion most wide applications would occur at such conditions.

To complete this review, I am a bit surprised that the authors do not mention any other application of their work but qubits analysis and processing. It could be wise to mention that increasing the $Q \times F$ product by minimum a factor of 10 would also have some repercussion on signal processing device and systems.

Reviewer #2 (Remarks to the Author):

The manuscript describes the optimization for quantum acoustodynamic applications of a High-overtone Bulk Acoustic wave Resonator (HBAR) based on an epitaxial NbN/AlN/GaN transducer with top Cr/Al electrodes grown on top of a SiC substrate. The purpose of the authors is to maximize both the quality factor of individual resonances and at the same time maximize the acoustic power located in the substrate. The results obtained by the authors are very impressive and this manuscript manages to answer some questions, or at least to give some pieces of answers to these questions, which have arisen in the acoustic resonators community in the last decade or two:

- The first question pertains to whether single crystal materials offer better performances than poly-crystalline ones, as far as acoustic resonators are concerned. This claim has been in the air since a long time: some industrial companies have even been founded on this basis in the field of bulk acoustic wave resonators in the last few years. It has however not clearly been evidenced experimentally. I personally think this manuscript gives some hints why this is true, especially Extended Data Fig. 8. This figure highlights that $Q \cdot f$ factors at room temperature remain in the range of 10^{14} Hz, or below, even for a perfect material. This is however also the case for poly-crystalline materials. When going towards cryogenic temperatures, however, the experimental results of the paper clearly show that single crystal material outperform poly-crystalline materials. So the answer to the above-mentioned question could be simply "single crystal materials offer better performance than poly-crystalline ones only at cryogenic temperatures". It was perhaps not the purpose of the authors to show this, but I want to thank the authors for having helped me (at least) to come to this conclusion.

- The second interesting point is that initially HBARs were fabricated for providing high quality factor resonators to be inserted into oscillator loops. This means that somehow, the electromechanical coupling factor of individual resonances need to be maintained to a minimum level. Indirectly, this requires somehow confining acoustic waves in the transducer, hence limiting power in the high quality substrate. Therefore, quality factors are ultimately traded-off for electrical resonance strength. On the contrary, the authors want the resonator to act as a phonon source, hence they do no longer require to compromise quality factor for signal strength. This helps them widely increase device performances, especially thanks to the exquisite choice of materials they discuss in the manuscript. This shows a different perspective.

So to summarize, I found the manuscript well written, very interesting, and it opens very large perspectives, even beyond the scope of the study.

I would however have some suggestions for the authors to improve their manuscript:

1) Line 3 of the section "Epitaxial HBARs with acoustic impedance matching": the resonance frequency is given as $f_r = \omega_r / 2\pi = (2r+1) v_r / t_r$. The authors use therefore the index "r" both for the order of the resonance and the indication of which material they are referring to. It would perhaps be clearer to replace this by $f_n = \omega_n / 2\pi = (2n+1) v_r / t_r$, as they do a few lines later when defining the frequencies of the substrate.

2) The authors only present and discuss resonators measurements at 7.2 K. Since they are not so many reports of HBARs characterized at such low temperatures, it would be interesting also to have an evaluation of the quality factors at room temperature, for comparison with other works. Clearly, the quality factors will be orders of magnitude worse than at 7 K, it would personally help me positioning the devices proposed by the authors with respect to other devices reported in the literature.

3) In the same order of ideas, Figure 4(a) presents Q.f. products reported in the literature for various types of transducers and substrates. However, the values reported are for temperature ranges in the 0.02-300 K range. So one can clearly not compare a quality factor at 300 K with one obtained at 20 mK. Could the authors add some way of identifying the temperature at which these values were obtained? For example adding a color code (red = 300 K, blue = 20 mK), or circling around a group of values obtained below 10 K and another obtained above 200 K?

4) In the supplementary material, the authors identify parasitic contributions at 1.5 and 2 GHz, which they attribute to the excitation of surface acoustic waves (SAW) or Sezawa waves. To confirm this, the authors have implemented some interdigitated transducers with varying electrodes pitches. Indeed, waves with wavelengths in the 3-5 μm range seem to exhibit resonances corresponding to the features seen on the electric response of HBARs. But this is not a clear proof that these surface waves are responsible for them. Especially, why would surface waves with wavelengths in this narrow range only be excited? In principle, surface waves with any wavelength could be excited, and exhibit contributions at any frequency. Since the transducer structure is relatively complex, couldn't the features visible on the electric response of HBARs also originate from additional modes generated in the transducer, aside from the main longitudinal bulk modes. For example due to the excitation of shear waves that are expected roughly at a frequency half of the one of longitudinal waves (i.e. in this particular 1.5-2 GHz region)?

5) The finite element simulations of Figure 1(a) and Extended Data 7(c) show that some acoustic leakage occurs laterally out of the resonator (which by the way are circular electrodes in these figures, not squares as mentioned in the "methods" text). This seems to mean that the resonator does not operate in energy trapping condition, which should normally limit strongly the quality factors. Isn't there a discrepancy with experimental measurements?

6) In a few parts of the text, the authors mention that a proper mounting of the resonator could further

decrease anchor losses. If the energy trapping condition is fulfilled, anchor losses are in principle non-existing, so this should no longer be a concern.

7) Final point: in the "Measurement and simulations of the epi-HBAR" section, could the authors indicate the frequency sampling they used in their measurements and the IF settings? Since they measure extremely large quality factors, these two settings may critically affect the values obtained.

Reviewer #3 (Remarks to the Author):

Overall the authors present an informative and interesting paper using all epi materials which have well matched acoustic impedances to create a high-Q, miniature, electrically transduced acoustic storage element for quantum information storage. The authors clearly demonstrate the benefits of employing layers with closely matched acoustic impedances and epitaxial growth in achieving long phonon relaxation times. While the authors have done a thorough job detailing their work and experiments, the comments below would help to clarify some aspects of the work and the conclusions drawn. I recommend the paper be published and hope the authors will consider addressing the items below.

1) The impedances reported in Fig. 2d indicate approximately 3x difference between Q_s and Q_p . The differences in Q_s and Q_p reported in Fig. 3a seem to be much smaller. Can the authors account for this difference? Is it from extracting electrical parasitics such as R_s ?

2) The authors should clarify if the Q values reported include the electrical parasitic elements R_s and R_f ? If not, can the authors discuss why it would be appropriate to remove these? Especially since the NbN bottom electrode, which has the advantage of excellent impedance matching, has the disadvantage of large resistivity, and thus large R_s , when compared to other electrode metals. This loss will be present unless thicker NbN is utilized which would likely result in higher acoustic damping. The loss is also clearly visible in Fig. 3 (e), (f), and (g). The impact of this potential source of loss should be acknowledged.

3) It would be more informative to show $|Z_{11}|$ or $|Y_{11}|$ in Fig. 3 (e), (f) and (g). This would show Q_s , Q_p , electromechanical coupling, and the impedance ratio, as opposed to how well the particular device is matched to 50 ohms. It would also demonstrate a much higher fidelity match between the model and the measured data.

4) In the figure 1 caption the last line should read "Note that (d) and (e) are do not have the same Y-axis scaling."

A point by point response to reviewers' comments for NCOMMS-19-36347-T:

Reviewer #1 (Remarks to the Author):

Abstract : the authors use several times vague terms like low, good, smooth, which to my own do not contribute positively to the paper quality. Although often read even in high quality papers, i would suggest to avoid using such terms without a quantification just as the authors do for phonon lifetime.

We appreciate the constructive comments. In the revised version, we have used terminology that is more precise and added numerical quantification wherever possible or appropriate. Changes are highlighted in the marked-up document.

The sentence "to achieve strong coupling and reduce..." is the type of example i would take care to avoid - strong coupling does not mean a thing, as it refers to nothing. Quartz is less coupled than LiTaO3 but the coupling can be considered strong for given cuts compared to other. Also it may useful to to write "to reduce" or "reduced" to the best of my English basis.

In light of this comment, we see the confusion the original sentence caused, and have amended the manuscript. In the introductory paragraph in question, we are discussing quantum electrodynamics as a forerunner to quantum acoustodynamics. In the sentence mentioned by the reviewer, we are discussing the strong coupling regime between photons and qubits (not the coupling strength of a particular cut of piezoelectric material). Ref. 1 in the previous sentence defines the strong coupling regime (interaction rate between the photon and qubit should be greater than dissipation rate). We have modified this sentence to clarify this point and fix grammatical issues; however, we do not think that a quantification for the general concept of strong coupling is appropriate here.

"significantly reducing footprint, power consumption, and complexity.": compared to what ?

The sentence has been modified to include "...as compared to off-chip macroscale phonon sources such as optomechanically transduced quartz resonators...."

Page 2, top, HBAR have been studied for the mentioned application but i do not know an effective industrial application of such devices which remains marginal in the corresponding industry (although extremely exciting).

We agree with this general sentiment: HBARs have not found widespread commercial success in the RF, sensing, or spectroscopy industries. Various practical reasons for this, which are not within the scope of this manuscript, are addressed in the cited references. We only wish to point out to the unfamiliar reader that HBARs have been studied in the past for various conventional applications. We believe that for integrated quantum engineering applications, the emphasis on high performance will make HBARs appealing and relevant.

Again, i am extremely sensitive to the mention of "g=high" Q factors without precising the values and the operating frequency. Classical BAW exhibit Q factors of about 2 millions at 5 MHz for instance whereas Q factors of HBARs may reach 40000 at 2 GHz using single crystals with direct electrical coupling, see for instance Pijolat & al publications (which is not mentioned in the reference list)."exquisitely narrow linewidths" means with Q factors exceeding the usual values for acoustic based devices (and in parenthesis, give a Q.F product estimate/ $Q.F \gg 10^{14}$ which is a representative value published by Lakin in early 90's).

In the Summary and Introduction, we have now added more direct quantitative comparisons between our work (both cryogenic and room temperature results) with appropriate representative values published in the literature. For the sake of space and conciseness, many prior studies using HBARs have been cited only in the Extended Data document. Some work by Pijolat et al was already included in the survey of prior work and has been cited as Ref. 25 in the Extended Data document.

Better than low-loss substrate, the authors should mentioned substrates with minimal viscoelastic losses as i understand it is actually what is looked for.

We have included language in the introduction to clarify that low-loss indicates low intrinsic phonon losses in this context, in order to not confuse with dielectric or resistive losses familiar to some readers.

The last paragraph of the introduction is an example of wording that does not contribute to science and reads like opinions as there is no quantification of $Q \times$ frequency and relaxation delay reference to which the authors results can be compared. It would be helpful to give some figures allowing for fixing the comparison, to evaluate the gain of $Q \times$ frequency product (never talk about Q factor without mentioning the operating frequency) and to objectively appreciate the author contribution.

We have now included text in the introduction highlighting our fQ results quantitatively to allow objective comparison. Q factors are now always mentioned with the corresponding frequency. In many cases, fQ values are quoted directly.

on the same page (bottom) i read again a mention to a good acoustic matching, whereas a simple reference to "acoustic matching" is clear and sufficient : layers are matched together as reflection coefficient is lower than for instance 1%.

We have removed the qualitative descriptors and instead, specified the Fresnel coefficients where appropriate in order to clarify and quantify the extent of acoustic impedance matching (or mismatch).

Q values of 10 millions are mentioned without mention to the frequency, it must be added: 10 GHz as the authors are focusing on the 3rd harmonics of the piezoelectric layer fundamental mode. I stop commenting these flaws as there is a lot to edit all along the paper.

All Q values in the text now include the corresponding mode frequency or frequency range.

On fig.4, reference is done to Lithium Niobate. Some precision would be welcome when showing such a graph without clear reference to previous work, which is in definitive difficult to interpret and to verify.

Since the fQ result for this lithium niobate data in Fig4 are not on par with our measurements, we have removed the explicit mention in the figure. We have retained the data since they do represent bulk crystal overtone resonators of interest, in important RF frequency bands. The lithium niobate data are from Y. Yang, R. Lu, T. Manzanque, and S. Gong, IEEE International Frequency Control Symposium, 2018, and are cited in the Extended Data document. The bulk quartz optomechanical resonators explicitly marked in Fig 4 are cited in the caption.

I want to mention that the results of the authors are remarkable and that they must present them at best, with a very care to avoid diffuse mentions and comparative adjectives without reference. Probably it would be of interest to mention the actual $Q \times f$ values they obtained at room temperature, as from my opinion most wide applications would occur at such conditions.

To complete this review, i am a bit surprised that the authors do not mention any other application of their work but qubits analysis and processing. It could be wise to mention that increasing the $Q \times F$ product by minimum a factor of 10 would also have some repercussion on signal processing device and systems.

We have included room temperature data in the Extended Data document. In addition, we have specified room temperature fQ values in the summary and introduction of the main paper. While the majority of this paper is deliberately focused on QAD systems, we have now added some text at the end of the paper discussing the potential implications for conventional signal processing systems. We certainly agree that the high fQ product seen in our work has positive repercussions for conventional signal processing systems. This will be a focus in our future work, but is well beyond the scope and size restrictions of this manuscript.

We want to thank the reviewer for affirming the remarkable nature of our results. We greatly appreciate the constructive criticism and feedback from the reviewer, and acknowledge that we needed improvements in the original manuscript to present our results in the best light. The review comments emphasize the need for more precise language in our paper. To that end, we have modified the manuscript at various points in order to provide a quantitative description of our work that can be quickly appreciated by an expert in the field. This revision has been somewhat tempered with the need to present a compelling narrative that is accessible to a reader not familiar with the field-specific jargon and quantifiable figures of merit. We believe that the revised manuscript strikes the right balance.

Reviewer #2 (Remarks to the Author):

The manuscript describes the optimization for quantum acoustodynamic applications of a High-overtone Bulk Acoustic wave Resonator (HBAR) based on an epitaxial NbN/AlN/GaN transducer with top Cr/Al electrodes grown on top of a SiC substrate. The purpose of the authors is to maximize both the quality factor of individual resonances and at the same time maximize the acoustic power located in the substrate. The results obtained by the authors are very impressive and this manuscript manages to answer some questions, or at least to give some pieces of answers to these questions, which have arisen in the acoustic resonators community in the last decade or two:

- The first question pertains to whether single crystal materials offer better performances than poly-crystalline ones, as far as acoustic resonators are concerned. This claim has been in the air since a long time: some industrial companies have even been founded on this basis in the field of bulk acoustic wave resonators in the last few years. It has however not clearly been evidenced experimentally. I personally think this manuscript gives some hints why this is true, especially Extended Data Fig. 8. This figure highlights that $Q \cdot f$ factors at room temperature remain in the range of 10^{14} Hz, or below, even for a perfect material. This is however also the case for poly-crystalline materials. When going towards cryogenic temperatures, however, the experimental results of the paper clearly show that single crystal material outperform poly-crystalline materials. So the answer to the above-mentioned question could be simply “single crystal materials offer better performance than poly-crystalline ones only at cryogenic temperatures”. It was perhaps not the purpose of the authors to show this, but I want to thank the authors for having helped me (at least) to come to this conclusion.

- The second interesting point is that initially HBARs were fabricated for providing high quality factor resonators to be inserted into oscillator loops. This means that somehow, the electromechanical coupling factor of individual resonances need to be maintained to a minimum level. Indirectly, this requires somehow confining acoustic waves in the transducer, hence limiting power in the high quality substrate. Therefore, quality factors are ultimately traded-off for electrical resonance strength. On the contrary, the authors want the resonator to act as a phonon source, hence they do no longer require to compromise quality factor for signal strength. This helps them widely increase device performances, especially thanks to the exquisite choice of materials they discuss in the manuscript. This shows a different perspective.

So to summarize, I found the manuscript well written, very interesting, and it opens very large perspectives, even beyond the scope of the study.

We appreciate such a positive summary of our work. As the reviewer points out, reducing internal loss in resonators has been an ongoing effort in the acoustic resonators community for years. We can make two rather broad observations about the pace of improvement. First, the maturity of materials and technology has often limited performance in the past. Secondly, some applications (especially commercial ones) have not required such high performance, removing the crucial motivation behind this research. As we look towards applications in quantum engineering, with different and more stringent requirements than conventional microacoustics, we believe we have both the motivation and the means to make significant improvements. While this work reports exceptional results, we firmly believe that performance can improve further. Even though this particular paper is more focused on low temperature applications, this research can positively influence applications at room temperature (and potentially higher).

I would however have some suggestions for the authors to improve their manuscript:

1) Line 3 of the section “Epitaxial HBARs with acoustic impedance matching”: the resonance frequency is given as $f_r = \omega_r / 2\pi = (2r+1) v_r / t_r$. The authors use therefore the index “r” both for the order of the resonance and the indication of which material they are referring to. It would perhaps be clearer to replace this by $f_n = \omega_n / 2\pi = (2n+1) v_r / t_r$, as they do a few lines later when defining the frequencies of the substrate.

Thank you for pointing out this oversight. In the revised manuscript, we use the index ‘n’ to refer to transducer mode number, and (as before) subscript ‘r’ to refer to the piezoelectric material.

2) The authors only present and discuss resonators measurements at 7.2 K. Since there are not so many reports of HBARs characterized at such low temperatures, it would be interesting also to have an evaluation of the quality factors at room temperature, for comparison with other works. Clearly, the quality factors will be orders of magnitude worse than at 7 K, it would personally help me positioning the devices proposed by the authors with respect to other devices reported in the literature.

Since the focus of this paper is on QAD applications of epi-HBARs, we prefer to present detailed analyses of data at 7.2 K in the main manuscript. However, multiple reviewers make the valid point that room temperature quality factors will be useful for comparison with other works. As such, we have now included room temperature data for the epi-HBAR as Extended Data Fig.6. We also mention key performance data at room temperature in the introduction and summary.

3) In the same order of ideas, Figure 4(a) presents Q.f. products reported in the literature for various types of transducers and substrates. However, the values reported are for temperature ranges in the 0.02-300 K range. So one can clearly not compare a quality factor at 300 K with one obtained at 20 mK. Could the authors add some way of identifying the temperature at which these values were obtained? For example adding a color code (red = 300 K, blue = 20 mK), or circling around a group of values obtained below 10 K and another obtained above 200 K?

All data in Fig. 4 (our epi-HBAR data as well as data reported in literature) already use a color fill (scaled from light blue (0 K) to bright red (300 K)) to identify the measurement temperature. We feel it is important to show our measured results in comparison with the full range of reported data, and not just the best-reported values. To simplify comparison in specific temperature ranges, we present a subsection of the literature survey in Extended Figure 11, showing all data below 10 K and all data at room temperature in two separate panels. Extended Figure 11 more clearly demonstrates that the epi-HBARs in this work have higher $f \times Q$ values at room temperature. Similarly, the epi-HBARs have higher $f \times Q$ values than almost all but the best macroscale bulk devices (with optomechanical actuation) at comparable cryogenic temperatures.

4) In the supplementary material, the authors identify parasitic contributions at 1.5 and 2 GHz, which they attribute to the excitation of surface acoustic waves (SAW) or Sezawa waves. To confirm this, the authors have implemented some interdigitated transducers with varying electrodes pitches. Indeed, waves with wavelengths in the 3-5 μm range seem to exhibit resonances corresponding to the features seen on the electric response of HBARs. But this is not a clear proof that these surface waves are responsible for them. Especially, why would surface waves with wavelengths in this narrow range only be excited? In principle, surface waves with any wavelength could be excited, and exhibit contributions at any frequency. Since the transducer structure is relatively complex, couldn't the features visible on the electric response of HBARs also originate from additional modes generated in the transducer, aside from the main longitudinal bulk modes. For example due to the excitation of shear waves that are expected roughly at a frequency half of the one of longitudinal waves (i.e. in this particular 1.5-2 GHz region)?

The main reason for including these data and simulations is to highlight the difference between an epi-HBAR (with a bottom electrode) and a control structure without a bottom electrode. The latter structure does not practically generate or trap longitudinal cavity phonons.

It is true that the features seen in the low frequency range could be due to other modes such as shear modes, and that in principle, surface waves of any wavelength can be excited. However, given the specific dimensions and layered structure of the GaN/NbN/SiC, SAW dispersion analyses indicate that the strongest piezoelectric coupling to surface acoustic waves is achieved for frequencies below 2.5 GHz. We acknowledge that this is not clear proof that these specific SAW modes are responsible for the parasitic low frequency modes, but think it is highly likely to be the case, based on both simulation and the control experiments previously discussed. We have modified the main manuscript and Extended Data to remove ambiguity, being careful not to claim that SAWs are solely responsible for the parasitic modes. If this frequency range is of interest to a potential application, well-designed simulations and experiments can unambiguously identify and eliminate the spurious modes.

5) The finite element simulations of Figure 1(a) and Extended Data 7(c) show that some acoustic leakage occurs laterally out of the resonator (which by the way are circular electrodes in these figures, not squares as mentioned in the "methods" text). This seems to mean that the resonator does not operate in energy trapping condition, which should normally limit strongly the quality factors. Isn't there a discrepancy with experimental measurements?

Due to the structure of the epi-HBAR, the energy trapping conditions are only valid for waves travelling in the vertical direction. Laterally actuated surface waves (based on the e_{31} piezoelectric coefficient of GaN and any small fringe fields along the surface) could leak out of the structure. The simulations use circular electrodes due to computational constraints: axisymmetric finite element simulations can be carried out with a higher mesh density than a full 3D simulation. We have carried out full 3D simulations using square electrodes for limited frequencies with qualitatively similar results. The extended data figure caption now includes a note clarifying the use of circular electrodes in simulation. A complete experimental and finite element analysis of electrode shape, size, and apodization dependence on HBAR performance is beyond the scope of this work, but is planned as future research in the near term.

6) In a few parts of the text, the authors mention that a proper mounting of the resonator could further decrease anchor losses. If the energy trapping condition is fulfilled, anchor losses are in principle non-existing, so this should no longer be a concern.

We have clarified the text to state unambiguously that if the energy trapping conditions were ideal, anchor losses would be eliminated. In practice, we believe it is possible that the presence of non-ideal conditions (thermal grease on the bottom of the substrate, wire-bonds too close to the HBAR, etc.) could detract from the ideal trapping conditions and influence the net damping, especially if all other loss mechanisms are low. However, since such effects are not systematically investigated at this point, we remove any speculative discussion from the revised manuscript.

7) Final point: in the "Measurement and simulations of the epi-HBAR" section, could the authors indicate the frequency sampling they used in their measurements and the IF settings? Since they measure extremely large quality factors, these two settings may critically affect the values obtained.

To ensure that data are not under-sampled and that extracted Q values converge, RF datasets were acquired as a number of sequentially acquired data-files. The spans of each individual data-file were between 250 kHz and 250 MHz. Each data file contains 32001 points (limited by the data buffer length of the vector network analyzer) at an IF acquisition speed of 1 kHz. Variations in this procedure (e.g. using acquisition speeds as slow as 10 Hz or averaging multiple scans), as well as analytical data interpolation and curve fitting for selected representative modes confirm that Q values had converged. A brief version of this clarification is added to the methods section.

Reviewer #3 (Remarks to the Author):

Overall the authors present an informative and interesting paper using all epi materials which have well matched acoustic impedances to create a high-Q, miniature, electrically transduced acoustic storage element for quantum information storage. The authors clearly demonstrate the benefits of employing layers with closely matched acoustic impedances and epitaxial growth in achieving long phonon relaxation times. While the authors have done a thorough job detailing their work and experiments, the comments below would help to clarify some aspects of the work and the conclusions drawn. I recommend the paper be published and hope the authors will consider addressing the items below.

We would like to thank the reviewer for the positive review comments and recommendation. We have addressed the specific comments in detail below.

1) The impedances reported in Fig. 2d indicate approximately 3x difference between Q_s and Q_p . The differences in Q_s and Q_p reported in Fig. 3a seem to be much smaller. Can the authors account for this difference? Is it from extracting electrical parasitics such as R_s ?

The values of Q_s and Q_p depend on the sharpness of the peaks. The series and parallel peaks shown in 2(d) are similar in their linewidth, with the parallel peaks observed to be only slightly sharper, (i.e., not a $3\times$ difference). This is reflected accurately in the Q values from measurements shown in Fig. 3(a). We do not remove the effect of electrical parasitics such as R_s for extracting Q_s and Q_p . However, it is important to reiterate that modeled Q_{BVD} considers only the mechanical branch of the model, and implicitly ignores the effect of parasitics ($Q_{BVD} = \omega_m L_m / R_m = 1 / \omega_m C_m R_m$). That is, Q_{BVD} is the only value where parasitics are removed, all other reported Q and fQ values in the paper are directly measured values 'loaded' with parasitics. As mentioned in the Methods section, there is close correspondence between measured quality factors (Q_s and Q_p) and modeled quality factors (Q_{BVD}).

2) The authors should clarify if the Q values reported include the electrical parasitic elements R_s and R_f ? If not, can the authors discuss why it would be appropriate to remove these? Especially since the NbN bottom electrode, which has the advantage of excellent impedance matching, has the disadvantage of large resistivity, and thus large R_s , when compared to other electrode metals. This loss will be present unless thicker NbN is utilized which would likely result in higher acoustic damping. The loss is also clearly visible in Fig. 3 (e), (f), and (g). The impact of this potential source of loss should be acknowledged.

As mentioned in the prior comment, we do not remove the effect of electrical parasitics such as R_s and R_f for the measured values of Q_s and Q_p (Fig. 3(a)). As the reviewer correctly points out, this loss is clearly visible in Fig 3(e)-(g). Ideally, we would want to optimize electrode thickness for both lower electrical and acoustic matching and loss. Another advantage of using NbN electrodes is the potential to operate in the superconducting regime, which would eliminate the electrical series resistance R_s . This point is now explicitly mentioned in the measurement section of Methods.

3) It would be more informative to show $|Z_{11}|$ or $|Y_{11}|$ in Fig. 3 (e), (f) and (g). This would show Q_s , Q_p , electromechanical coupling, and the impedance ratio, as opposed to how well the particular device is matched to 50 ohms. It would also demonstrate a much higher fidelity match between the model and the measured data.

Generally, we prefer to present S_{11} data since this is the raw data obtained from the network analyzer without downstream data processing. The impedance and admittance are derived from the raw S_{11} data (without any parasitic extraction or removal), and are now included in the Extended Data. Extended Data Fig. 10 shows the $|Y_{11}|$ and $|Z_{11}|$ for modes 164, 476, and 529 (corresponding to the S_{11} data from Fig. 3(e)-(g)). The BVD modeled $|Y_{11}|$ and $|Z_{11}|$ are included in the corresponding panels. In all cases, the impedance includes the effect of all parasitic elements. As reinforced by this figure, the modeled data are well matched with the measured data.

4) In the figure 1 caption the last line should read "Note that (d) and (e) are do not have the same Y-axis scaling." Thank you. We have corrected the typographical error.

REVIEWERS' COMMENTS:

Reviewer #1 (Remarks to the Author):

This reviewer provided confidential remarks to editor with no objection to publication.

Reviewer #2 (Remarks to the Author):

I want to thank the authors for their excellent work: the performances of their HBARs are outstanding, even at room temperature. They answered convincingly the comments and questions of the reviewers.

I would just indicate a minor issue overlooked during the first review and appearing also in the changes brought to the revised manuscript: f_xQ factors have the unit of a frequency. While the authors indicated these units in the Introduction and in Figures, they are missing in later sections of the text (e.g. line 5 of the "QUALITY FACTOR AND PHONON RELAXATION TIME" section, " $f_xQ \rightarrow 10^{17}$ " is written instead of " $f_xQ \rightarrow 10^{17} \text{ Hz}$ "). Please, correct this throughout the whole text.

Reviewer #3 (Remarks to the Author):

Overall the authors present an informative and interesting paper using all epi deposited materials on a single crystal Silicon Carbide substrate, where all the layers have well matched acoustic impedances, to create a high-Q, miniature, electrically transduced acoustic storage element for quantum information storage. The authors clearly demonstrate the benefits of employing layers with closely matched acoustic impedances and epitaxial growth in achieving long phonon relaxation times. The authors present a comprehensive comparison with prior art including sputtered materials and optically transduced single crystals. The frequency times quality factor (fQ) products reported demonstrate record performance levels for electrically transduced acoustic resonators both at room temperature and at cryogenic temperatures. Given the importance of fQ product as a metric for acoustic resonators, the results reported are of high significance. The performance in terms of quality factors and relaxation times are presented in easily readable charts both at cryogenic and room temperatures and properly compared to prior art. The authors have diligently addressed the review feedback, which has significantly improved the communication of these exciting results. The reported results will be of high interest to both the acoustic resonator and quantum information research communities and provides new information on the achievable performance in highly overmoded bulk acoustic resonator devices. I recommend the paper be published based on the outstanding work reported by the authors.

A response to reviewers' comments for NCOMMS-19-36347A:

Reviewer #1 (Remarks to the Author):

This reviewer provided confidential remarks to editor with no objection to publication.

Reviewer #2 (Remarks to the Author):

I want to thank the authors for their excellent work: the performances of their HBARs are outstanding, even at room temperature. They answered convincingly the comments and questions of the reviewers.

I would just indicate a minor issue overlooked during the first review and appearing also in the changes brought to the revised manuscript: fQ factors have the unit of a frequency. While the authors indicated these units in the Introduction and in Figures, they are missing in later sections of the text (e.g. line 5 of the "QUALITY FACTOR AND PHONON RELAXATION TIME" section, " $fQ \rightarrow 10^{17}$ " is written instead of " $fQ \rightarrow 10^{17}$ Hz"). Please, correct this throughout the whole text.

Thank you for catching the oversight. The 'Hz' unit was missing in two places; this has now been corrected.

Reviewer #3 (Remarks to the Author):

Overall the authors present an informative and interesting paper using all epi deposited materials on a single crystal Silicon Carbide substrate, where all the layers have well matched acoustic impedances, to create a high-Q, miniature, electrically transduced acoustic storage element for quantum information storage. The authors clearly demonstrate the benefits of employing layers with closely matched acoustic impedances and epitaxial growth in achieving long phonon relaxation times. The authors present a comprehensive comparison with prior art including sputtered materials and optically transduced single crystals. The frequency times quality factor (fQ) products reported demonstrate record performance levels for electrically transduced acoustic resonators both at room temperature and at cryogenic temperatures. Given the importance of fQ product as a metric for acoustic resonators, the results reported are of high significance. The performance in terms of quality factors and relaxation times are presented in easily readable charts both at cryogenic and room temperatures and properly compared to prior art. The authors have diligently addressed the review feedback, which has significantly improved the communication of these exciting results. The reported results will be of high interest to both the acoustic resonator and quantum information research communities and provides new information on the achievable performance in highly overmoded bulk acoustic resonator devices. I recommend the paper be published based on the outstanding work reported by the authors.

Summary Response

We want to thank all reviewers for their careful and critical evaluation of our work over two review cycles. The final revised version is significantly improved as a direct result. We believe this version will do a better job of clearly communicating our results and their significance, not just to researchers working on acoustic resonators and quantum devices, but also to the larger research community.